# Cross-Sectional Association of Dietary Patterns and Supplement Intake with Presence and Gray-Scale Median of Carotid Plaques—A Comparison between Women and Men in the Population-Based Hamburg City Health Study

**DOI:** 10.3390/nu15061468

**Published:** 2023-03-18

**Authors:** Julia Maria Assies, Martje Dorothea Sältz, Frederik Peters, Christian-Alexander Behrendt, Annika Jagodzinski, Elina Larissa Petersen, Ines Schäfer, Raphael Twerenbold, Stefan Blankenberg, David Leander Rimmele, Götz Thomalla, Nataliya Makarova, Birgit-Christiane Zyriax

**Affiliations:** 1Midwifery Science—Health Care Research and Prevention, Research Group Preventive Medicine and Nutrition, Institute for Health Service Research in Dermatology and Nursing (IVDP), University Medical Center Hamburg-Eppendorf, Martinistraße 52, W26, 20246 Hamburg, Germany; julia.assies@googlemail.com (J.M.A.); martje@saeltz.de (M.D.S.); b.zyriax@uke.de (B.-C.Z.); 2Hamburg Cancer Registry, 20097 Hamburg, Germany; 3Population Health Research Department, University Heart and Vascular Center, 20246 Hamburg, Germany; 4Lohfert & Lohfert Working Group, 20148 Hamburg, Germany; 5Department of Cardiology, University Heart and Vascular Center, 20246 Hamburg, Germany; 6German Center for Cardiovascular Research (DZHK), Partner Site Hamburg/Kiel/Luebeck, 20246 Hamburg, Germany; 7Department of Neurology, University Medical Centre Hamburg-Eppendorf, Martinistraße 52, 20246 Hamburg, Germany

**Keywords:** dietary patterns, supplements, carotid artery disease, cardiovascular disease, peripheral artery disease, carotid plaques, GSM, prevention

## Abstract

This population-based cross-sectional cohort study investigated the association of the Mediterranean and DASH (Dietary Approach to Stop Hypertension) diet as well as supplement intake with gray-scale median (GSM) and the presence of carotid plaques comparing women and men. Low GSM is associated with plaque vulnerability. Ten thousand participants of the Hamburg City Health Study aged 45–74 underwent carotid ultrasound examination. We analyzed plaque presence in all participants plus GSM in those having plaques (*n* = 2163). Dietary patterns and supplement intake were assessed via a food frequency questionnaire. Multiple linear and logistic regression models were used to assess associations between dietary patterns, supplement intake and GSM plus plaque presence. Linear regressions showed an association between higher GSM and folate intake only in men (+9.12, 95% CI (1.37, 16.86), *p* = 0.021). High compared to intermediate adherence to the DASH diet was associated with higher odds for carotid plaques (OR = 1.18, 95% CI (1.02, 1.36), *p* = 0.027, adjusted). Odds for plaque presence were higher for men, older age, low education, hypertension, hyperlipidemia and smoking. In this study, the intake of most supplements, as well as DASH or Mediterranean diet, was not significantly associated with GSM for women or men. Future research is needed to clarify the influence, especially of the folate intake and DASH diet, on the presence and vulnerability of plaques.

## 1. Introduction

Atherosclerotic cardiovascular disease (CVD) is widespread and is a leading cause of morbidity and mortality worldwide [1,2]. Atherosclerosis refers to a slowly progressive process of plaque formation in the vessel wall. Plaque rupture, platelet activation and, consequently, secondary thrombosis may occur during the progression of the disease [3,4,5,6]. Thereby, the risk of cardio- and cerebrovascular events is increased. Ischemic strokes are caused by the rupture of plaques in the carotids in about 15% of cases [7,8]. The stability and vulnerability of plaques have a major impact on the risk of plaque rupture [9]. Stable plaques consist of a high amount of fibrous tissue and calcification. Unstable plaques, in contrast, are rupture-prone due to high lipid content, an oftentimes necrotic core and intra-plaque hemorrhage [10,11].

Measurement of carotid intima-media thickness (cIMT) and its progression is an established and widely used prognostic biomarker for future CVD events [12,13]. However, it does not provide any information about plaque composition and, thus, the vulnerability of plaques. Therefore, the measurement of plaque gray-scale median (GSM) may improve the detection of vulnerable plaques. GSM provides additional information on plaque morphology due to the measurement of densitometry of the plaque [14,15].

The previous literature has demonstrated that GSM is a suitable measurement to quantify and assess the vulnerability of carotid plaques based on their echogenicity on B-mode ultrasonography [14,16,17]. GSM correlates with histopathological findings in patients after carotid endarterectomy and thus reflects the composition of plaques [10,18,19,20,21,22]. More precisely, high GSM values correlate with predominantly echogenic, stable plaques with a higher grade of calcification and fibrosis, whereas low GSM values are associated with echolucent, vulnerable plaques [11,14]. Low GSM values in carotid plaques are associated with an increased risk for CVD events, especially ischemic strokes [23,24,25].

GSM and cIMT are associated with different risk factors. While cIMT correlates with traditional risk factors such as hypertension and smoking status, GSM correlates with other traditional risk factors like dyslipidemia as well as with markers of inflammation and oxidative stress [26,27]. This suggests that cIMT and GSM may depict different aspects of atherosclerosis, with GSM relating more to metabolic aspects [28]. In addition to GSM, the presence of carotid plaque is associated with the incidence of CVD events [29,30] and is further an established ultrasound surrogate of CVD [31,32]. Nutritional aspects are known to play a relevant role in the development of atherosclerosis and in the formation of plaques [33]. Hence, it is of great interest how both can be influenced through diet.

CVD may be prevented in up to 90% of cases by a healthy lifestyle [34]. Numerous studies investigating the association between dietary patterns and CVD have included Dietary Approaches to Stop Hypertension (DASH) diet and the Mediterranean diet. Both adherence to the DASH diet and the Mediterranean diet are usually higher in women than in men [35,36,37]. Mediterranean diet has shown a primary prevention effect on CVD events as well as a tendency to slow down carotid plaque progression [38,39,40,41,42,43,44]. However, data on the association with GSM has been missing until now, and even the association between the Mediterranean diet and cIMT remains to be confirmed [45]. DASH diet is associated with fewer CVD events and lower cIMT values, while data regarding the association with GSM or plaque presence is not available [35,46,47].

Furthermore, dietary supplement intake is widespread in the general population [48]; for example, more than half of US adults take at least one supplement daily [49,50]. For Germany, the EPIC-Heidelberg cohort has shown an increasing prevalence of up to about 45% for vitamin/mineral supplement intake in a follow-up reassessment (2004–2006) [51]. The EPIC-Heidelberg cohort and many other studies have also revealed that women, in particular, are more likely to take supplements compared to men [49,51,52,53,54]. The main reasons for intake are general health and well-being and filling nutrient gaps [55].

According to previous studies, the associations between dietary supplements and CVD or cIMT remain unclear. Some data on B vitamins exist, especially for folic acid supplementation, which appears to be associated with benefits for CVD and, in particular, stroke risk [56,57,58,59]. Studies investigating associations between dietary supplement intake with GSM or plaque presence are lacking.

This study, therefore, aimed to examine associations between the dietary patterns Mediterranean diet and the DASH diet as well as dietary supplements (specifically multivitamins, multiminerals, calcium, magnesium, vitamin B and folate) and (a) the presence or (b) GSM of carotid plaques as predictors of CVD in women and men.

## 2. Materials and Methods

### 2.1. Study Population and Study Design

This study is part of the Hamburg City Health Study (HCHS). HCHS is a prospective, single-center, population-based cohort study. It aims to identify risk and prognostic factors of main chronic diseases. Participants must be inhabitants of Hamburg, Germany, at the time of enrollment, aged 45–74 years and must provide sufficient language skills for participating in the study. Participants are chosen randomly via the registration office. They sign an informed consent and undergo an extensive baseline evaluation. Detailed information on the HCHS has been published separately [60]. For this study, data from the first sub-cohort (*n* = 10,000) was used. Data acquisition took place between 8 February 2016–30 November 2018.

### 2.2. Ultrasound Images

B-mode duplex sonography was performed by trained study assistants using a Siemens SC2000^®^ Ultrasound System and a 7.5 Mhz broadband linear transducer. Measurement of the cIMT was performed three times. The carotid bulb, common carotid artery and internal and external carotid artery were then scanned for plaques using the longitudinal view of carotid artery. A plaque was defined as a local cIMT ≥ 1.5 mm.

### 2.3. Gray-Scale Median

Carotid ultrasound scans were saved in DICOM (digital imaging and communications in medicine) format after performing the sonography. In the next step, echogenicity of carotid plaques was analyzed using software that was specifically written for this project’s purpose, based on the open-source project JS Paint [61,62]. Plaques were segmented manually by outlining the plaques using the computer mouse. One additional marker was drawn in the vessel lumen, and a second in the adventitia. Each plaque was segmented twice by different operators to minimize interobserver reliability. Interobserver reliability was determined based on a random sub-sample of 135 (5%) participants that were evaluated by all observers. Remeasurements of outliers were performed. Images were saved as portable network graphics (PNG) files after segmentation. Next, image brightness was normalized using the vessel lumen as the reference structure for darkness (GSM = 0) and the adventitia as the reference structure for brightness (GSM = 190). Both grayscale values were chosen based on the existing literature [63]. In general, GSM values range from 0, indicating total black, to 255, indicating total white. Noise reduction and cropping of the images were performed automatically. Finally, minimum, maximum, mean and median grayscale values were calculated and output in a comma-separated values (CSV) file. Primary outcome of the present study was the mean value over all individual echogenicity measurements as numerical variable.

### 2.4. Questionnaires and Dietary Scores

Dietary habits and intake of nutrition supplements were assessed in questionnaires. For dietary intake, the food frequency questionnaire (version 2, FFQ2) developed for the European Perspective Investigation into Cancer and Nutrition (EPIC) study was used [64]. It samples information on frequency and portion size of 102 food items consumed during the previous year. Information was collected and analyzed in terms of energy intake, food groups and nutrients.

The validated German translation of the Mediterranean Diet Adherence Score (MEDAS) was used for evaluating adherence to a Mediterranean diet [65]. It contains twelve questions on food items and two questions on food habits (Appendix A). For each item, a score of 0 indicates a non-adherence, whereas a score of 1 indicates adherence. Finally, the score was grouped by quantiles into the categories 0–3, 4, 5 and 6+.

Adherence to the Dietary Approaches to Stop Hypertension (DASH) diet was assessed using a scoring system adapted from Folsom et al. [66]. The score includes ten items on consumption of grains, vegetables, fruits, dairy, meat/poultry/fish, nuts/seeds/legumes and sweets (obtained from raw data) and average daily intake of nutrients (saturated fat, fat, sodium) (Appendix A). Each item was scored from 0 to 1. Finally, the score was grouped by quantiles into the categories 0–3.5, 3.6–4.5, 4.6–5.0 and 5.1+.

The FFQ2 continued to ask about the use of dietary supplements for at least one month in the last twelve months, specifically multipreparations (multivitamin or multimineral preparations or both) or 14 single and simple combination preparations, as well as nine natural health products. For this study, data on multivitamin and multimineral preparations as well as calcium, magnesium, vitamin B complex and folic acid, were included.

### 2.5. Statistical Analysis

In the descriptive analysis, continuous data are presented as the median and interquartile range (IQR), and categorical data as absolute numbers and percentages.

Multiple linear regressions were used to assess the association between echogenicity and dietary and supplement intake, i.e., nutritional supplements, DASH diet, MEDAS within GSM-sub-cohort (*n* = 2163). All models were estimated separately for males and females and adjusted for not performing any sports (examined as ‘never performing sports except for cycling or walking’), age, socioeconomic status index (including education, profession, salary), body mass index (BMI), smoking status, energy intake (kcal), dyslipidemia, hypertension, diabetes mellitus, myocardial infarction, heart failure, atrial fibrillation, history of stroke or transient ischemic attack (TIA), peripheral arterial disease, estimated glomerular filtration rate (eGFR), lipid-lowering drugs, antihypertensive medication, antidiabetic medication, use of antiplatelets. Central results were presented as betas with 95% confidence intervals. We did not adjust for multiple comparisons. We imputed missing values by multivariate imputation by chained equations separately for twenty copies of the data with ten iterations. Subsequently, estimates were averaged, and standard errors were adjusted using Rubin’s rules [67].

We performed additional analysis regarding the presence of at least one carotid plaque using multiple logistic regressions within a full cohort of 10,000 participants. For the full-adjusted model, age, sex, education, body-mass index, diabetes mellitus, arterial hypertension, hyperlipidemia, smoking status, heart failure, atrial fibrillation, myocardial infarction, stroke and sports were used for adjustment. Education was divided into three categories (low, medium and high) based on the International Standard Classification of Education (ISCED 1011).

Statistical significance was defined as an α = 0.05. We adhered to the Strengthening the Reporting of Observational Studies in Epidemiology (STROBE) statement [68]. All analyses were performed in R version 4.0.3.

## 3. Results

### 3.1. Baseline Characteristics of GSM-Sub-Cohort

From the HCHS cohort of 10,000 participants, GSM was assessed for 2163 participants having at least one carotid plaque (Figure 1). The baseline characteristics of these participants, consisting of 921 (42.6%) women and 1242 (57.4%) men, are shown in Table 1. Here, the median age of women and men at recruitment was 68 (IQR (62, 73)) years. Obesity was found in 272 (21.9%) men and 187 (20.3%) women. Overall, 486 (22.5%) were current smokers. Of men, 397 (32.0%) were not performing any sports, whereas 249 (27.0%) women were not exercising.

Women reached higher MEDAS scores more often than men; women reached a score of 6+ points in 37.8% of the cases, whereas men reached a score of 6+ points in 16.1%. A similar trend holds true for the DASH score: 31.1% of women and 14.2% of men achieved a score of 5.1+ points. Men reached the largest distribution range at 0–3.5 points (31.8%) and 3.6–4.5 points (30.4%). In comparison, fewer women had low score values.

A total of 755 (34.9%) participants had an intake of any supplement. Intake was higher among women (43.8%) than men (23.8%). As Table 1 shows, for each of the examined supplements, intake was higher in women than in men, with the exception of multivitamins. Here, an equal supplementation distribution of 7.7% each for women and men was assessed.

Figure 2 shows the distribution of GSM levels separately for men (shown in blue) and women (shown in red). The median GSM was 56.50 with IQR between 46.00 and 68.50 for men and 55.80 with IQR between 44.25 and 70.33 for women.

### 3.2. Linear Regression of Nutrition Parameters and Examined Supplements with GSM in Women and Men

Table 2 shows the results of multivariate linear regression models of nutrition parameters, examined supplements and GSM in men and women of the GSM-sub-cohort (*n* = 2163). A significant correlation could only be found for folic acid intake in men (GSM 9.12 (95% CI (1.37, 16.86), *p* = 0.021).

A non-significant opposing trend was found in women with GSM of −2.50 (95% CI −9.31, 4.31), *p* = 0.472. No significant associations could be found between dietary patterns or intake of the other examined supplements and GSM.

### 3.3. Results of Logistic Regression Regarding the Presence of Carotid Plaques

The results of logistic regressions with multivariable adjustments as OR related to the reference category for the presence of at least one carotid plaque in full HCHS-sub-cohort, including 10,000 participants, are shown in Table 3. In all three adjusted logistic regression models, the odds for the presence of at least one carotid plaque were significantly higher among the categories men, older age, low education, arterial hypertension, hyperlipidemia and smoking status (Appendix A).

A high DASH score showed significantly increased odds for the presence of at least one plaque compared to intermediate score values in adjusted models (OR = 1.18, 95% CI (1.02, 1.36), *p* = 0.027).

In adjusted models, no significant association between MEDAS or any supplement intake and the presence of carotid plaque was found.

## 4. Discussion

GSM was not associated with Mediterranean or DASH nutritional patterns and most supplements in an elderly German population. Folic acid intake was significantly associated with higher GSM only in men. A high DASH score was significantly associated with increased odds for the presence of carotid plaques compared to intermediate score values. However, in all other fully adjusted analyses, no significant associations were found between DASH/Mediterranean diet and plaque presence.

This study is the first to investigate associations between GSM and the Mediterranean Diet or DASH diet as well as the supplements examined in this study, plus the relation between the presence of carotid plaques with the DASH diet or supplement intake. There are only a few studies that have investigated plaque prevalence and MEDAS.

The study’s baseline data fit with the demographics of previous studies, which have also shown that both following healthy dietary patterns—measured by high adherence scores—and taking supplements are more prevalent among women [35,36,37,49,53,69,70].

The significantly increased GSM in men taking folic acid should be considered with caution because only 30 men (2.4%) supplemented folic acid. Future studies should investigate the effect of folic acid on plaque vulnerability. In addition to that, the clinical implication should be mentioned. If the observed evidence of a 9.12 increased GSM by folic acid intake (95% CI (1.37, 16.86), *p* = 0.021) is not coincidental, this positive effect, however, is not necessarily clinically relevant. However, three reviews revealed a reduced stroke risk for folic acid supplementation and, thus, beneficial effects for stroke prevention [57,58,59]. Again, further studies are necessary to determine which GSM changes are clinically relevant to outcomes related to CVD, e.g., ischemic stroke. Thus, the findings probably exist due to confounders like traditional cardiovascular risk factors considering that supplement users tend to have more healthy habits than non-users [55].

Several studies have shown that the presence of carotid plaques is particularly associated with older age, male sex [71] and smoking [72], but also linked to diseases such as hypercholesterolemia [31], hypertension, diabetes mellitus [73,74] and cardiac disease [75]. Our findings are in line with previous studies that revealed the following associations: In adjusted regression models, the odds of having at least one plaque significantly increased in men, older age, low education, arterial hypertension, hyperlipidemia and smoking status. Evidence for correlations between supplement intake or DASH diet with plaque presence is missing in the existing literature. For any supplement intake, the odds of carotid plaque presence were lower, although no significant trend was observed after adjustment.

Contrary to our expectations, we have found a significant association between high DASH scores and a more frequent occurrence of carotid plaques in adjusted models. In contrast, Fung et al. showed that adherence to the DASH diet is associated with a reduced risk of CVD events such as stroke [46]. The reason for our findings could be that people having cardiovascular diseases are more willing to follow healthy nutrition recommendations. Likewise, individuals who have received nutritional counseling cause of their CVD are more likely to report healthy nutrition in questionnaires (recall/reporting bias).

We found an absence of proof regarding the association between MEDAS or supplement intake and the presence of carotid plaque. Previous studies confirm that there may be no association between MEDAS and the presence of carotid plaques. For example, neither Gardener et al. in the Northern Manhattan Study (NOMAS) [38] nor Mateo-Gallego et al. in the Aragon Workers’ Health Study (AWHS) [41] observed an association between the Mediterranean diet and plaque presence. Jimenez-Torres et al. also did not find any effect of the Mediterranean diet on the number of carotid plaques [39]. In contrast, a Croatian study in a population of HIV-infected patients found that lower adherence to the Mediterranean diet was associated with increased odds of subclinical atherosclerosis defined as cIMT ≥ 0.9 mm or ≥ 1 carotid plaque [76].

Although no clinically relevant association between the Mediterranean/DASH diet or supplement intake and GSM has been found, some studies have shown associations between these lifestyle adjustments and the CVD predictor cIMT. For example, Maddock et al. describe significantly lower cIMT for greater adherence to the DASH diet [35].

Because GSM and cIMT may be associated with different risk factors [26,27] and represent different aspects of atherosclerosis [28], it is worth doubting whether GSM is an appropriate parameter for detecting associations with dietary adjustments. Perhaps other methods are more useful for investigating associations and, finally, causal influences on clinical outcomes related to diet or supplements. For example, using a juxtaluminal black area (JBA) instead of GSM could provide even more information [22]. While the GSM value is based on the echolucency measurement of the whole plaque, JBA focuses on a low GSM plaque area near the vessel lumen. Salem et al. found a stronger association between histological findings and JBA than with GSM [21].

In summary, further research regarding the relationship between GSM and the presence of carotid artery plaques with nutrition patterns or supplement intake is needed.

### Strengths and Limitations

The present study consists of an exceptionally large sample size of 2163 participants within the GSM-sub-cohort and 10,000 participants in an additional analysis with the presence of at least one plaque. Almost no exclusion criteria (only insufficient German language skills and incapability to travel to the study center and to cooperate in the investigations) and random invitations via the registration office are used for the selection of study participants for HCHS. Still, selection bias cannot be excluded for certain. HCHS participants tend to be more health-conscious and educated, showing fewer cardiovascular risk factors than the general German population [77]. Furthermore, the HCHS study population consists of middle-aged individuals living in Hamburg, so generalizations to other age groups and individuals living in rural areas should not be made without careful consideration.

Being a cross-sectional analysis, no causal conclusions can be made. Data on dietary parameters were collected by self-reporting in questionnaires, so there is a risk of reporting and recall bias. In addition, no data were collected on the dose of the supplements nor on the continuity or duration of intake.

Furthermore, adjustments for multiple comparisons were not performed. This could lead to dismissing the null hypothesis hastily, especially in consideration of the wide variety of supplements.

Another limitation could be our grouping of the dietary scores in the GSM regression models (MEDAS 4/5/6+ points, DASH 3.6–4.5, 4.6–5, 5.1+) since differences in adherence between the groups are small. Comparison of, for instance, the highest tertial of adherence vs. the lowest tertial of adherence might have been more informative. In addition, adherence to the Mediterranean diet was low in our northern German participants. Another dietary pattern, e.g., an anti-inflammatory or Nordic diet, could have shown higher prevalence rates and thus more information.

Additionally, in the present study, GSM measurement was performed based on 2D ultrasound scans and thus cannot present information on the whole plaque as 3D files may have done.

Lastly, multiple trained operators drew in the plaques for the GSM determination. This leaves room for intra-observer and inter-observer variability. Plus, the reference values for normalization of the image brightness also had to be drawn in. This can lead to a bias in true GSM values if, for instance, an expert draws in an area that is too dark for the adventitia, the normalization thus becoming incorrect [78].

## 5. Conclusions

The current study found no clinically relevant significant associations between adherence to the DASH/Mediterranean diet or supplement intake and the GSM of carotid plaques. There may be an association between higher GSM and folate intake in men, but further studies are needed to confirm this association and clinical relevance.

High compared to intermediate adherence to the DASH diet was associated with higher odds for carotid plaque presence.

Further research is needed to examine whether nutrition patterns or supplement intake—particularly DASH diet and folate intake—are associated with plaque presence or GSM.

## Figures and Tables

**Figure 1 nutrients-15-01468-f001:**
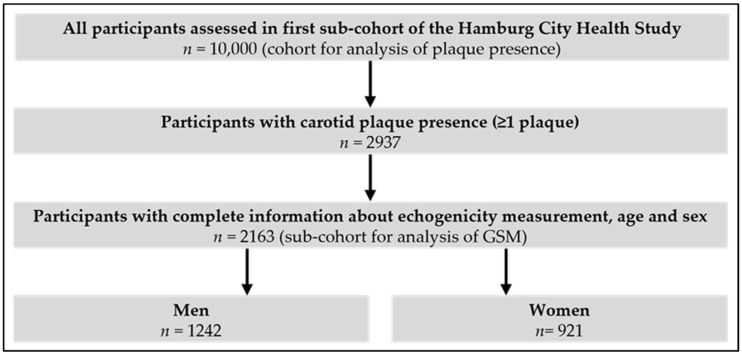
Flow Chart for assessment of echogenicity measurement in participants of HCHS.

**Figure 2 nutrients-15-01468-f002:**
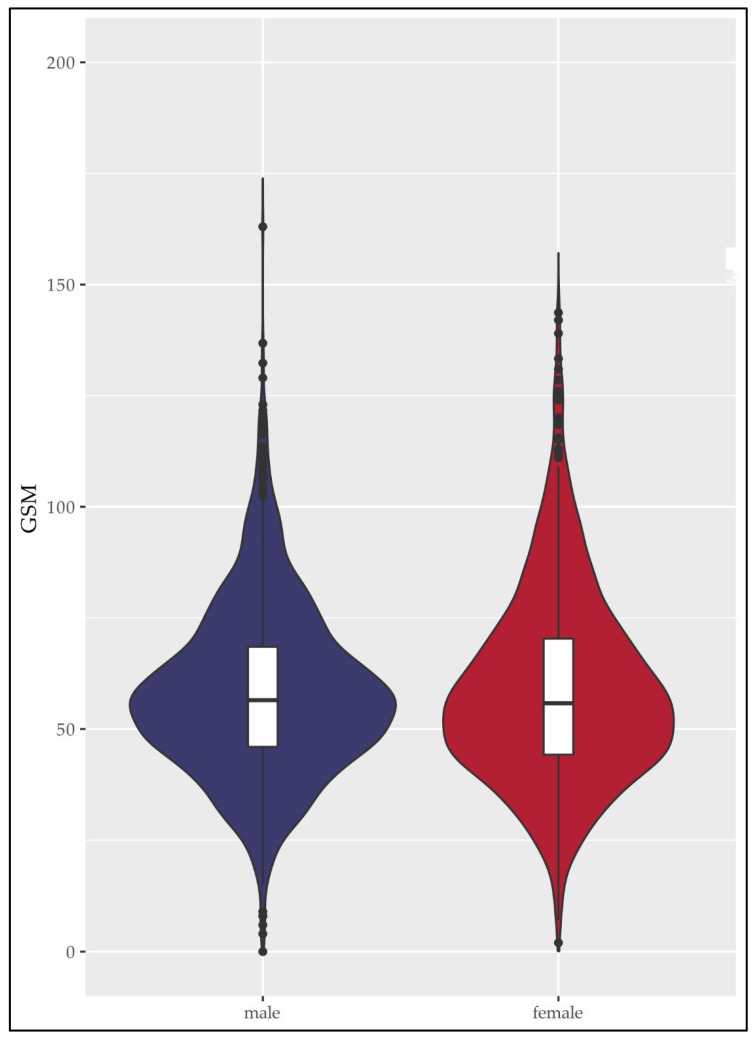
Distribution of gray-scale median (GSM) in women (red) and men (blue). *n* = 2163. Abbreviations: GSM, gray-scale median.

**Table 1 nutrients-15-01468-t001:** Baseline characteristics of the study population of GSM-sub-cohort.

	Overall	Men	Women	*p*-Value
n (%)	2163 (100)	1242 (57, 4)	921 (42, 6)	
Age in years (median [IQR])	68 (62, 73)	68 (62, 73)	68 (62, 72)	0.44
SES index (median [IQR])	12.30 (9.97, 16.00)	13.30 (10.30, 16.70)	11.40 (9.40, 14.20)	<0.001
NA	1307 (60.4)	653 (52.6)	654 (71)	
BMI (kg/m^2^) (%)				<0.001
NA	130 (6.0)	67 (5.4)	63 (6.8)	
Normal weight (BMI 18.5–24.9 kg/m^2^)	677 (31.3)	322 (25.9)	355 (38.5)	
Obesity (BMI ≥ 30 kg/m^2^)	459 (21.2)	272 (21.9)	187 (20.3)	
Overweight (BMI 25–29.9 kg/m^2^)	883 (40.8)	576 (46.4)	307 (33.3)	
Underweight (BMI < 18.5 kg/m^2^)	14 (0.6)	5 (0.4)	9 (1.0)	
Smoking status (%)	486 (22.5)	282 (22.7)	204 (22.1)	0.697
NA	11 (0.5)	5 (0.4)	6 (0.7)	
Not performing any sports (%)	646 (29.9)	397 (32.0)	249 (27.0)	0.047
NA	180 (8.3)	100 (8.1)	80 (8.7)	
Total energy intake, kcal (median [IQR])	2009.48 (1613.75, 2564.23)	2311.52 (1853.48, 2837.52)	1729.92 (1419.01, 2101.23)	<0.001
NA	242 (11.2)	141 (11.4)	101 (11)	
MEDAS score (%)				<0.001
NA	244 (11.3)	142 (11.4)	102 (11.1)	
0–3 points	576 (26.6)	453 (36.5)	123 (13.4)	
4 points	439 (20.3)	272 (21.9)	167 (18.1)	
5 points	356 (16.5)	175 (14.1)	181 (19.7)	
6+ points	548 (25.3)	200 (16.1)	348 (37.8)	
DASH score (%)				<0.001
NA	244 (11.3)	142 (11.4)	102 (11.1)	
0–3.5 points	489 (22.6)	395 (31.8)	94 (10.2)	
3.6–4.5 points	643 (29.7)	377 (30.4)	266 (28.9)	
4.6–5.0 points	325 (15.0)	152 (12.2)	173 (18.8)	
5.1+ points	462 (21.4)	176 (14.2)	286 (31.1)	
Any supplement intake (%)	755 (34.9)	352 (28.3)	403 (43.8)	<0.001
NA	177 (8.2)	100 (8.1)	77 (8.4)	
Multivitamins (%)	167 (7.7)	96 (7.7)	71 (7.7)	1
Multiminerals (%)	180 (8.3)	78 (6.3)	102 (11.1)	<0.001
Calcium (%)	127 (5.9)	68 (5.5)	59 (6.4)	0.413
Magnesium (%)	382 (17.7)	179 (14.4)	203 (22.0)	<0.001
Vitamin B (%)	110 (5.1)	45 (3.6)	65 (7.1)	<0.001
Folate (%)	76 (3.5)	30 (2.4)	46 (5.0)	0.002
Hyperlipidemia (%)	744 (34.4)	498 (40.1)	246 (26.7)	<0.001
NA	100 (4.6)	53 (4.3)	47 (5.1)	
Arterial hypertension (%)	1644 (76.0)	974 (78.4)	670 (72.7)	<0.001
NA	62 (2.9)	44 (3.5)	18 (2.0)	
Diabetes mellitus (%)	256 (11.8)	171 (13.8)	85 (9.2)	0.002
NA	118 (5.5)	59 (4.8)	59 (6.4)	
Prior MI (%)	119 (5.5)	103 (8.3)	16 (1.7)	<0.001
NA	17 (0.8)	10 (0.8)	7 (0.8)	
Heart failure (%)	174 (8.0)	115 (9.3)	59 (6.4)	0.014
NA	20 (0.9)	15 (1.2)	5 (0.5)	
Atrial fibrillation (%)	183 (8.5)	120 (9.7)	63 (6.8)	0.014
NA	199 (9.2)	101 (8.1)	98 (10.6)	
Prior stroke (%)	100 (4.6)	65 (5.2)	35 (3.8)	0.121
NA	17 (0.8)	7 (0.6)	10 (1.1)	
PAD (ABI < 0.9) (%)	255 (11.8)	140 (11.3)	115 (12.5)	0.308
NA	1140 (52.7)	645 (51.9)	495 (53.7)	
GFR (median [IQR])	87.20 (78.40, 93.20)	88.90 (81.20, 94.60)	84.80 (76.30, 90.30)	<0.001
NA	211 (9.8)	105 (8.5)	106 (11.5)	
Lipid-lowering drugs (%)	617 (28.5)	413 (33.3)	204 (22.1)	<0.001
NA	60 (2.8)	39 (3.1)	21 (2.3)	
Antihypertensives (%)	1000 (46.2)	598 (48.1)	402 (43.6)	0.035
NA	60 (2.8)	39 (3.1)	21 (2.3)	
Antidiabetics (%)	171 (7.9)	119 (9.6)	52 (5.6)	0.001
NA	60 (2.8)	39 (3.1)	21 (2.3)	
Antiplatelets (%)	600 (27.7)	409 (32.9)	191 (20.7)	<0.001
NA	60 (2.8)	39 (3.1)	21 (2.3)	

This table shows baseline characteristics related to participants with assessed GSM (*n* = 2163). Abbreviations: IQR, interquartile range; SES, socioeconomic status; NA, not available (missings with respect to line above); BMI, body mass index (calculated as weight in kilograms divided by height in meters squared); MEDAS, Mediterranean Diet Adherence Score; DASH, Dietary Approach to Stop Hypertension; MI, myocardial infarction; PAD, peripheral artery disease; ABI, ankle-brachial-pressure-Index; GFR, glomerular filtration rate.

**Table 2 nutrients-15-01468-t002:** Linear regression models for Outcome GSM (0 to 255) in men and women.

Model Parameters	Men		Women	
	GSM [95% CI] ^1^	*p*-Value	GSM [95% CI] ^1^	*p*-Value
MEDAS 4 points ^2^	−0.14 (−3.06, 2.79)	0.927	−2.82 (−7.78, 2.15)	0.267
MEDAS 5 points ^2^	−1.10 (−4.45, 2.25)	0.521	−2.02 (−6.78, 2.74)	0.406
MEDAS 6+ points ^2^	−1.53 (−4.71, 1.66)	0.347	−1.88 (−6.19, 2.43)	0.393
DASH 3.6–4.5 points ^3^	−1.27 (−4.09, 1.55)	0.378	−1.69 (−6.57, 3.19)	0.497
DASH 4.6–5.0 points ^3^	−0.95 (−4.66, 2.75)	0.614	−3.76 (−9.06, 1.53)	0.164
DASH 5.1+ points ^3^	−1.45 (−4.86, 1.96)	0.406	−0.33 (−5.22, 4.57)	0.896
Any supplement intake ^4^	−0.69 (−4.12, 2.74)	0.693	−1.06 (−4.78, 2.67)	0.578
Multivitamins ^4^	−2.67 (−7.94, 2.61)	0.322	−2.03 (−8.53, 4.47)	0.540
Multiminerals ^4^	1.52 (−4.11, 7.15)	0.597	−0.87 (−6.53, 4.79)	0.763
Calcium ^4^	−0.14 (−5.57, 5.29)	0.960	2.57 (−3.61, 8.76)	0.415
Magnesium ^4^	−0.01 (−4.20, 4.18)	0.995	−0.19 (−4.43, 4.05)	0.930
Vitamin B ^4^	−2.19 (−8.59, 4.21)	0.502	3.93 (−1.93, 9.79)	0.189
Folate ^4^	9.12 (1.37, 16.86)	0.021	−2.50 (−9.31, 4.31)	0.472

This table shows results of linear regression models regarding GSM related to participants with assessed GSM presented as betas (*n* = 2163). ^1^ Model adjusted for: age, socioeconomic status, not doing sport, BMI, smoking status, energy intake (kcal), dyslipidemia, hypertension, diabetes mellitus, myocardial infarction, heart failure, atrial fibrillation, history of stroke or TIA, peripheral arterial disease, eGFR, lipid-lowering drugs, antihypertensive medication, antidiabetic medication, antiplatelets. ^2^ Reference category: <4 points. ^3^ Reference category: <3.6 points. ^4^ Reference category: No supplement intake. Abbreviations: GSM, gray-scale median; CI, confidence interval; MEDAS, Mediterranean Diet Adherence Score; DASH, Dietary Approach to Stop Hypertension; BMI, body-mass-index; TIA, transient ischemic attack; eGFR, estimated glomerular filtration rate.

**Table 3 nutrients-15-01468-t003:** Logistic regression models regarding the presence of carotid plaques.

Characteristics	OR (95% CI)	*p*-Value
High MEDAS vs. medium MEDAS	1.07 (0.92, 1.24)	0.367
Low MEDAS vs. medium MEDAS	0.86 (0.75, 1.00)	0.052
High DASH vs. medium DASH	1.18 (1.02, 1.36)	0.027
Low DASH vs. medium DASH	0.95 (0.82, 1.10)	0.469
Supplement intake yes vs. no	0.96 (0.85, 1.08)	0.490

This table shows results of additional analyses of logistic regression models regarding the presence of carotid plaques in full HCHS-sub-cohort, including 10,000 participants. All models are adjusted for age, sex, education, body-mass index, diabetes mellitus, arterial hypertension, hyperlipidemia, smoking status, heart failure, atrial fibrillation, myocardial infarction, stroke and sports. Abbreviations: OR, odds ratio; CI, confidence interval; MEDAS, Mediterranean Diet Adherence Score; DASH, Dietary Approach to Stop Hypertension.

## Data Availability

Data for this study as well as statistical analyses and R code are available for purposes of review, transparency and comprehensibility.

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
