# Peer review of "Cross-Sectional Association of Dietary Patterns and Supplement Intake with Presence and Gray-Scale Median of Carotid Plaques—A Comparison between Women and Men in the Population-Based Hamburg City Health Study"

_nutrients, 2023, doi:10.3390/nu15061468_

Round 1
Reviewer 1 Report
This population-based cross-sectional cohort study from Assies et al. enrolled 10,000 participants from the Hamburg City Health Study to evaluate the association of dietary patterns and supplement intake with presence and gray-scale median of carotid plaques. Dietary Approach to Stop Hypertension and Mediterranean diets were evaluated and the supplements analyzed were multivitamins, multiminerals, calcium, magnesium, vitamin B and folate.
The manuscript is well designed and presented. The limitations are indicated at the end of the manuscript.
It deserves be accept for publication after few adjustment.
Few adjustment are suggested:
ABSTRACT
Line 28: change "This population-based cross-sectional cohort study investigated the association of both established dietary patterns" to "This population-based cross-sectional cohort study investigated the association of TWO established dietary patterns."
Line 28-29: I suggest to indicate the names of the diets target of this study.
Line 40: change 'DASH/Mediterranean diet' to 'DASH or Mediterranean diet'
Materials and Methods
Materials and Methods should be 2 instead of 1
Results
Line 195: Please provide the age interval after the mean age
Figure 2. Please provide the meaning of GSM in the figure legend.
Author Response
To the Editor of the Journal
Nutrients (Special Issue "Nutrition and Specific
Diseases by Women during the Life Course")
Hamburg, 03/15/2023
"Cross-sectional association of dietary patterns and supplement intake with presence and gray-scale median of carotid plaques – a comparison between women and men in the population-based Hamburg City Health Study" (Manuscript ID: nutrients-2257370)
by Julia Maria Assies, Martje Dorothea Sältz, Frederik Peters, Christian-Alexander Behrendt, Annika Jagodzinski, Elina Larissa Petersen, Ines Schäfer, Raphael Twerenbold, Stefan Blankenberg, David Leander Rimmele, Götz Thomalla, Nataliya Makarova and Birgit-Christiane Zyriax
Dear reviewers,
we’re glad for the possibility to re-submit our revised manuscript nutrients-2257370 “Cross-sectional association of dietary patterns and supplement intake with presence and gray-scale median of carotid plaques - a comparison between women and men in the population-based Hamburg City Health Study” for publication in Nutrients, Special Issue Nutrition and Specific Diseases by Women during the Life Course.
We thank you and the reviewers for your time and effort you dedicated to review our manuscript. We highly appreciated the valuable feedback on how to improve our manuscript. All of the recommendations were incorporated in the revised manuscript and point-by-point replies to the reviewers were phrased.
The revised manuscript is submitted with tracked changes as well as a “clean version”. Based on the comments, we
- intensively revised the abstract; among others we added information on the connection between GSM and plaque vulnerability and added additional significant subsidiary results
- added information on GSM in the introduction
- made several edits in the entire manuscript as well as the Graphical Abstract to clarify the distinction between the different sub-cohorts
- substituted Figure 3 with a concise table to reduce redundant information.
With these revisions, we hope to present an improved version of our manuscript.
Thank you again for consideration of our revised manuscript for publication.
Sincerely,
Nataliya Makarova for the authors

Reviewer 2 Report
This study aimed to examine associations between the dietary patterns (MEDA and DASH diet) and dietary supplements and (a) the presence or (b) GSM of carotid plaques as predictors of CVD in women and men.
There are several concerns about the design and presentation.
In the flow chart, we can see the authors only included participants at least with one carotid plaque, then they used logistic regressions to examine whether MEDA and DASH were linked to carotid plaque presence.
I think the authors should also include people without carotid plaque, then to see whether MEDA and DASH diet were related to presence or not carotid plaque.
The authors can also use people without carotid plaque as the reference group, and categorize number of carotid plaques, i.e., to treat number of carotid plaques as a multiple categories variable, then used multinomial logistic regression.
The presentations of the tables and figures are not clear.
In table 1, what did NA mean? Although the authors gave a notation, but it is not clear at all.
In Table 2, i.e., the linear regression models for GSM (0 to 255) in men and women. The authors said the "Reference category: <4 points. or Reference category: <3.6 points." This is confused. Since linear regression models were used, it means how GSM change with each one unit change of the exposure (e.g., one point change of MEDA). I am not clear why there is a Reference category here.
Further, in table 2, the 95%CI of the regression coefficients were rather wide. I suggest the authors consider each three points change of exposure to see its effect on GSM.
The three sub-figures in Figure 3 are redundant and not necessary. Except the results of MEDA, DASH and supplement, the other results were repeated in the three graphs. The authors can use a table to only include findings of MEDA, DASH and supplement.
Author Response

(The authors gave the same response as above.)

Reviewer 3 Report
This study is novel in the use of gray-scale median (GSM) of carotid plaques. Also, the study of supplements directly is uncommon. The dataset is an excellent one with which to undertake this analysis. The written English is perfect. Several aspects of these findings need to be clarified for the reader. There are actually two studies here, one of supplements and GSM and the other of diet and the presence of plaque. The excessive Figure 3 needs to be addressed. Other issues are described here.
Abstract
There are several problems in the Abstract that make the results unclear to the average reader.
-There are actually two study populations here. For the GSM, the study N was 2,163 subjects with plaque. For the DASH outcome, the population was the full 10,000. It took several readings for this to become clear. Would the authors please clarify these two study populations and outcomes.
-The connection between GSM and plaque vulnerability is needed in the Abstract. It would help explain that folate intake is associated with lower risk of vulnerable plaque.
-The fact that the folate result was not significant in women would useful to point out.
-Is the positive association between the DASH diet and plaque in accord with the hypothesis of the authors? I think not. An indication of this would be helpful to readers.
Introduction
-Line 59-60: This sentence is key for the novice to understand the entire article. A brief explanation of what grey-scale plaque is, would be helpful. It is not “superior” to plaque as much as it improves or supplements the information from plaque. If it was only superior, then why did the authors use the whole 10,00 population to find a benefit of the DASH diet? Please expand on this section to help introduce your audience to GSM.
-Line 71-72: Dyslipidemia is not a traditional risk factor? The authors may want to clarify this.
Results
-Line 248-250: With such limited significant results for diet and supplements, it might be interesting to include these positive, though previously well-known, results in the Abstract.
Figure 3 has several problems.
-It is more common practice to compare the first and third tertiles rather than the first and second and the second and third. The authors might add an extra line for this information.
-Also the graphics are not needed to repeat the relatively uninteresting results from the other controlling variables. A small table that summarizes the results from the two diets and the supplements would be sufficient and then a statement that in all three models, the following other variables were significant---. Please reduce these figures to a simple table.
Discussion
In the limitations, the authors need to acknowledge the fact that they did not adjust for multiple comparisons. In a study such as this in which the authors were surveying a wide variety of supplements for significant associations (some would say fishing), the lack of adjustment is unfortunate.
Author Response

(The authors gave the same response as above.)
